# Interleukin-6 and Melatonin as Predictors of Cognitive, Emotional and Functional Ageing of Older People

**DOI:** 10.3390/ijerph17103623

**Published:** 2020-05-21

**Authors:** Anna Kurowska, Iwona Bodys-Cupak, Magdalena Staszkiewicz, Joanna Szklarczyk, Joanna Zalewska-Puchała, Anna Kliś-Kalinowska, Marta Makara-Studzińska, Anna Majda

**Affiliations:** 1Laboratory of Theory and Fundamentals of Nursing, Institute of Nursing and Midwifery, Faculty of Health Sciences, Jagiellonian University Medical College, ul. Michałowskiego 12, 31-126 Krakow, Poland; i.bodys-cupak@uj.edu.pl (I.B.-C.); j.zalewska-puchala@uj.edu.pl (J.Z.-P.); a.klis-kalinowska@uj.edu.pl (A.K.-K.); anna.majda@uj.edu.pl (A.M.); 2Department of Clinical Nursing, Institute of Nursing and Midwifery, Faculty of Health Sciences, Jagiellonian University Medical College, ul. Kopernika 25, 31-501 Krakow, Poland; m.staszkiewicz@uj.edu.pl; 3Department of Medical Physiology, Faculty of Health Sciences, Jagiellonian University Medical College, ul. Michałowskiego 12, 31-126 Krakow, Poland; joannam.szklarczyk@uj.edu.pl; 4Department of Health Psychology, Institute of Nursing and Midwifery, Faculty of Health Sciences Jagiellonian University Medical College, ul. Kopernika 25, 31-501 Krakow, Poland; marta.makara-studzinska@uj.edu.pl

**Keywords:** melatonin, interleukin, ageing, functional skills, cognitive skills, emotional skills

## Abstract

Background: The ageing process causes a number of changes in the human immune and endocrine systems. The aim of this study was to assess the relationship between cognitive, emotional and functional skills as well lifestyle, versus selected biochemical indicators of the ageing process. Methods: The cross-sectional study was conducted in a group of 121 people aged 60–90 residing in the Lesser Poland voivodship. The study used standardized research tools including the Barthel scale, Instrumental Activities of Daily Living (IADL) scale, Mini-Mental State Examination (MMSE), Life Orientation Test (LOT-R) and inventory of health behaviors (IHB). In addition, the concentration of IL-6 and melatonin in the blood plasma was determined. Results: We determined the correlation between the level of IL-6 in a group of people over 75 years of age (requiring medical care), and results of the IADL scale. There was also a correlation between melatonin levels and the MMSE results in a group of people aged 60–75 who did not require constant medical care. Conclusions: IL-6 can be treated as a predictor of functional skills of people over 75 years of age, and melatonin can be perceived as a factor for recognizing cognitive impairment in elderly people who do not require constant medical assistance.

## 1. Introduction

Various steps are taken to ensure that older people remain independent. Some main factors that affect independence include cognitive, emotional and functional skills [1,2], the lack of which increases the sense of disability and powerlessness and increases the frequency of hospitalization and institutionalization into care and treatment facilities.

Human ageing is accompanied by changes in the immune system, including chronic subliminal inflammation which means a 2- to 4-fold increase in circulating pro-inflammatory cytokines. The most important of the cytokines is interleukin-6 (IL-6). It may show both pro-inflammatory properties (by stimulating antibody production and inducing acute inflammatory process) and anti-inflammatory properties (by blocking the synthesis of inflammatory cytokines) [3]. From the perspective of gerontology, interleukin-6 may contribute to the development of osteoporosis and induce development of anemia associated with chronic diseases by stimulating production of hepcidin by the liver cells. It is also thought to affect deposition of β-amyloid in cerebral vessels, thereby increasing the risk of Alzheimer’s disease. One can presume that it influences the accumulation of β-amyloid in brain vessels, thus increasing the risk of Alzheimer’s disease. Its importance in the pathophysiology of atherosclerosis and other cardiovascular diseases cannot be underestimated. In these patients, this is a risk factor for future disability development. Higher levels of IL-6 in healthy older people can be regarded as a marker of subclinical disease syndromes resulting in disability within a few years [3,4,5,6,7,8,9,10].

Oxidative stress and the pineal hormone (melatonin) are associated with pathogenesis of ageing and diseases associated with old age [11,12,13,14,15,16,17]. The ageing process is accompanied by increased accumulation of reactive oxygen and nitrogen (ROS: reactive oxygen species; RNS: reactive nitrogen species) produced in the mitochondria and lipid oxidation products (free radical ageing theory). Oxidative stress also accompanies many diseases including the cardiovascular system [18,19,20]. It is currently believed that the main effect of melatonin is to protect tissues from damage by oxidative stress. As a lipophilic substance, it has the ability to penetrate into the cells, protecting the mitochondria and the intracellular structures from destruction and the cell membrane from peroxidation. The antioxidant activity of melatonin relies on inactivating accumulated ROS and RNS as a result of insufficient tissue antioxidant capacity [21]. It acts directly by neutralizing free radicals, as well as indirectly as a powerful activator of antioxidant enzymes such as superoxide dismutase (SOD), catalase (CAT) or glutathione peroxidase (GPx) [22,23].

Based on the division of individual stages of old age proposed by the World Health Organization (WHO), respondents were assigned in this study to two age groups: 60–75 years old, which is considered early old age (elderly people) and 76–90 years old which is called the period of old age (senile people) [24].

While designing the study, the multidisciplinary research team hypothesized based on the available literature that there is a relationship between melatonin and interleukin-6 levels and the aging process.

The main aim of the study was to assess the relationship between cognitive, emotional and functional skills as well as lifestyle, and selected biochemical indicators of the ageing process in elderly people (60–75 years old) and senile people (76–90 years old).

The following specific objectives were identified in the study:Assessment of the relationship between cognitive, emotional and functional skills and lifestyle versus the level of the hormone melatonin in elderly people and senile people.Assessment of the relationship between cognitive, emotional and functional skills and lifestyle versus the level of pro-inflammatory cytokine interleukin-6 (IL-6) in elderly people and senile people.

## 2. Materials and Methods

### 2.1. Study Design

The cross-sectional study was conducted between 2018 and 2019, in a group of 121 people after obtaining the consent of 10 managers of facilities located in the lesser Poland voivodship. In 2018, funds were obtained under the statutory project number N43/DBS/000060. The facilities which participated in the study provided services and support to elderly and senile people which included care and treatment institutions (ZOL), daily houses of healthcare (DDOM), municipal daily houses of social care (MDDPS), senior activity centers (CAS) and activation clubs for seniors (KAS). The survey study combined with biochemical blood analysis presented in this paper consisted of four stages: (1) information, (2) interview with the study participant, (3) complementing the research tools with the study participant, (4) blood collection of the study participant. It was conducted by a multidisciplinary team of research and development staff (nurses, a laboratory worker a psychologist).

### 2.2. Procedure

The following criteria was required for inclusion in the study: (1) written consent to participate in the study, (2) absence of mental illness, Alzheimer’s disease or other severe neurodegenerative diseases, (3) absence of terminal cancer, (4) absence of visible infection features such as pneumonia and throat inflammation, (5) adequate preparation for the blood test which required 6–8 h of fasting and (6) no melatonin supplements. The participants of the study were provided with all necessary information about it, and were informed about the purpose of the study, voluntary participation in it, and the possibility of withdrawal at every stage of its conduct. Participants were not selected for convenience or through the user lists. Targeted selection of participants was used in the study. Potential participants were selected in consultation with the facility managers. The main criterion was informed consent to participate in the study, which was associated with absence of mental illness, Alzheimer’s disease or other severe neurodegenerative diseases which make it difficult to make an informed decision.

### 2.3. Ethical Considerations

The study was carried out after obtaining confirmation from the Bioethical Committee (Opinion No. 1072.6120.205.2018 of 20 September 2018). It was conducted and developed in accordance with the principles of good scientific practice, the Act of 10 May 2018 on the protection of personal data and the principles of the Helsinki Declaration. It was also carried out in accordance with the Regulation (EU) 2016/679 of the European Parliament and of the Council of 27 April 2016 on the protection of natural persons with regard to the processing of personal data and on the free movement of such data, and repealing Directive 95/46/EC (General Data Protection Regulation) [25,26].

### 2.4. Sampling

Data from the Central Statistical Office of Poland indicate that the number of old people (65+) in the lesser Poland voivodship in 2018 was 680,178 people [27]. Therefore, respondents participating in the study constituted 0.02% of the reference population. Analyzing the data only for the city of Krakow (from which over 80% of respondents came from), it can be seen that in 2018 the number of senile people registered in the city was 183,036. Therefore, the group participating in the survey constituted approximately 0.07% of the population of people over 65 in Krakow [28].

The study group (SG) consisted of women and men in elderly in two age groups (60–75 years old and 76–90 years old). These people underwent inpatient care in a care and treatment institution (ZOL) and temporarily in the daily houses of healthcare (DDOM) and required medical care. A total of 66 people met the inclusion criteria.

The elderly and senile women and men who lived at home, were active and benefited from the services of the MDDPS, CAS and KAS were in the comparative group (CG). There were 55 respondents in total in this group. The people who met the previously mentioned inclusion criteria did not require inpatient medical care.

The sociodemographic characteristics of the SG and CG are presented in Table 1.

### 2.5. Methods, Techniques and Research Tools

The study used the following research methods: (1) estimation method with standardized research tools in the form of test scales, (2) diagnostic survey method in the form of metrics and interviews, (3) statistical methods including techniques for measuring central tendency, measures of variability and statistical tests, (4) blood biochemistry by assessing interleukin-6 and melatonin in the venous plasma and (5) analysis of medical documentation available in medical facilities study to determine coexisting diseases.

Assessment of each patient in terms of functional skills took place on the basis of selected test scales. To assess skills in basic everyday activities (e.g., eating, moving, maintaining personal hygiene, using the toilet, bathing, dressing, going up and down the stairs) the Barthel scale (Barthel index) was used. According to this scale, the respondents were divided into three groups. The first group consisted of people who were defined as having a mild health condition or good functional condition (86–100 points). The second group has a medium level of health conditions/functional impairments (21–85 points) and the third group was defined as having a severe health condition/disability (0–20 points) [29,30,31,32].

To assess complex daily activities (instrumental functioning), Lawton’s Instrumental Activities of Daily Living (IADL) scale was used [29,33]. This made it possible to evaluate activities such as using the telephone, shopping, doing housework, cleaning and managing money.

Another research tool applied to assess the cognitive skills (cognitive function) of the older people was the Mini Mental State Examination (MMSE), which was developed by M.F. Folstein, S.E. Folstein and G. Fanjang. The tool was adapted to Polish conditions by J. Stańczak. MMSE assesses basic cognitive functions [34,35] and is a standardized research tool employed to screen dementia changes, while simultaneously assessing memory, counting, attention, concentration, visual–spatial coordination, orientation and language functions.

In this study, the Life Orientation Test (LOT-R) [36] was also implemented which was developed by M.F. Scheier, Ch. S. Carver and M.W. Bridges. R. Poprawa and Z. Juczyński carried out the adaptation of the research tool to Polish conditions. The LOT-R measures available optimism. One should note that optimistic people use emotion-based strategies such as acceptance or sense of humor more often in problematic situations [37,38]. Optimism is often treated as one of the most important personal resources conditioning human’s physical and psychological well-being. Therefore, it was used as a component in this study to analyze the assessment of emotional skills of older people.

The last standardized research tool used in the study was inventory of health behaviors (IHB) [36] developed by Z. Juczyński, which was applied to assess the lifestyle of older people. This research tool allowed us to measure the overall indicator of the intensity of health behavior. It consists of 24 statements that relate to a variety of health-related behaviors. This inventory is used to assess the intensity of preventive behaviors, proper eating habits and health practices, as well as positive mental attitudes.

The brief demographic questionnaire attached to the survey was specially developed for the study and allowed for gathering of information on sociodemographic variables such as age, gender, place of residence and coexisting diseases. The latter data was also confirmed during the analysis of medical documentation.

The study participants independently filled in three research tools (IADL, LOT-R and the brief demographic questionnaire). While completing the other two standardized research tools (MMSE and the Barthel scale), a member of the research team was present and asked questions to the subject. The MMSE and the Barthel scale are tools completed by a person with medical/ psychological education. The participant cannot objectively fill these tools in on their own.

### 2.6. Biochemical Blood Test

The collection of 5 mL of venous blood in vacuum tubes to determine the concentration of interleukin-6 (IL-6) and melatonin in plasma was one of the stages of the study. Venous blood was collected from each participant in the morning. Each tube contained K3EDTA (1.6 mg/mL blood) along with aprotinin (50 KIU/mL blood), which is an inhibitor of proteolytic enzymes. After thoroughly mixing the tubes, the collected material was placed in a centrifuge and centrifuged for 10 min at a speed of 3500 revolutions per minute (rpm). The centrifuged plasma was put in Eppendorfs (1.5 mL) and frozen at a temperature of −80 °C. The plasma was also stored at this temperature. After collection of the appropriate number of blood samples from participants, the tests were performed free of charge using commercial kits in a medical laboratory located at the University. The determinations were made on samples without hemolysis and lipemia. All measurements were carried out in accordance with the instructions of the manufacturer of the laboratory kits used in the study. Blood tests were performed by enzyme immunoassay method.

During the blood test, all precautions were taken and the principles of aseptic and antiseptic were followed. The study participants received details about the correct procedure after blood was taken. The subjects had the opportunity to become familiar with the results of the blood tests. Additional information was provided that the test results may be intended for research use only and not for use in diagnostic procedures.

### 2.7. Statistical Analysis

The study applied the following statistical methods: Chi-square test, Fisher’s exact test, Mann–Whitney U-test and Spearman rank correlation coefficient. The comparison of the values of qualitative variables in the groups was made implementing the Chi-square test (with Yates correction) or Fisher’s exact test where the low expected numbers appeared in the tables. The comparison of quantitative variable values in two groups was performed using the Mann–Whitney U-test. The correlations between quantitative variables were analyzed using the Spearman rank correlation coefficient. The analysis assumed the significance level α = 0.05. The collected research material was prepared in the R program, version 3.6.1 [39].

## 3. Results

The article shows statistically significant differences between the study and comparative groups. The study group had a smaller percentage of women and the respondents were older (Table 1).

People living at home, actively participating in classes organized by MDDPS, CAS and KAS (belonging to the comparative group) had better functional skills during basic daily activities which was indicated by a higher score on the Barthel scale. They also showed greater efficiency during the performance of complex daily activities, which was indicated by higher results obtained from the IADL scale, and also presented greater efficiency in the field of cognitive ability (shown with higher MMSE results) compared to the people in the study group. In addition, the respondents from the CG had more intense health behaviors and higher results in individual subscales of the IHB, including proper eating habits, preventive behaviors and a positive mental attitude compared to people undergoing inpatient care in ZOL and those in DDOM who required medical care (Table 2).

After analyzing the results of blood tests, it can be stated that in the study group the average level of melatonin was 436.85 ng/L, while in the comparative group the average was slightly higher (471.24 ng/L). The average concentration of interleukin-6 in the blood plasma of the study group was 22.29 pg/L and in the comparative group it was 17.07 pg/L. It should be noted that these groups did not differ significantly in the level of proinflammatory cytokine-interleukin-6 or the level of melatonin, which was demonstrated in our statistical analysis (*p* > 0.05) (Table 3).

The analysis of the Barthel scale results showed that in the study group the largest one consisted of people whose functional status could be defined as “medium level” (33 respondents, 50%) while in the comparative group, the vast majority were people whose functional status could be defined as “mild” (54 respondents, 98.18%). Most of the people in the study group presented cognitive impairment or reduced performance in the field of mental ability (55 of the people surveyed, 83.33%). In the comparative group, the majority of individuals who were evaluated had cognitive skills that were considered correct (39 people, 70.91%). It should be noted that in the study group the largest percentage of respondents were those with a high level of dispositional optimism—in other words, a high level of emotional skills (29 respondents, 43.94%). The comparative group was dominated by people with an average level of dispositional optimism (24 respondents, 43.64%). In both the study and comparative groups, the highest percentage were those with a high intensity rate of health behaviors (SG, 33 people, 50.00%; CG, 37 people, 67.27%) (Table 4).

The statistical analysis did not show any relationship between the level of interleukin-6 and melatonin versus functional, cognitive and emotional skills and the degree of health behavior intensity (results from standardized tools) among elderly respondents (60–75 years) belonging to the study group who were undergoing inpatient care in ZOL and DDOM (Table 5).

There was a relationship between the level of melatonin in elderly people (60–75 years) belonging to the control group and the results obtained from the MMSE scale. In the group of older people showing activity in everyday life, the MMSE scale results correlated significantly (*p* < 0.05) and negatively with the level of melatonin. In other words, the higher the level of this hormone, the lower the MMSE score (poorer cognitive skills). The relationship between the level of interleukin-6 and functional, cognitive and emotional skills and intensity of health behaviors was not found in the comparative group (Table 6).

A relationship between the level of interleukin-6 and melatonin in senile patients from the SG and the results obtained from the IADL scale and IHB was found. The IADL results correlated significantly (*p* < 0.05) and negatively with the level of interleukin-6, which meant that the higher the pro-inflammatory cytokine level, the lower the functional skills in performing complex daily activities (lower IADL results) were. It should be noted that the results of the subscale of proper eating habits correlated significantly (*p* < 0.05) and negatively with the level of melatonin. Higher levels of melatonin correlated with less proper eating habits in the group of people over 75 years old (Table 7).

The statistical analysis showed the relationship between the level of interleukin-6 and melatonin of people over 75 years of age belonging to the CG, and the results of the IADL scale, Barthel scale and LOT-R scale. The IADL scale results correlated significantly (*p* < 0.05) and negatively with the level of interleukin-6. The higher the level of this cytokine, the lower the functional skills in performing complex daily activities in a group of people using various forms of activity. The results of the Barthel scale, IADL scale and LOT-R scale correlated significantly (*p* < 0.05) and positively with the level of melatonin. The higher the level of this hormone, the better the functional skills (higher results of the Barthel scale), the greater the efficiency in performing complex daily activities (higher IADL scale results) and the greater the tendency for optimism (higher LOT-R scale results) (Table 8).

## 4. Discussion

Currently, in Poland and other European Union countries, the rapidly ageing population is being observed, which is largely associated with the extension of life expectancy and low fertility levels. The United Nations (UN) predicts that by 2030 the proportion of older people (over 65 years old) in Europe will be 23.8%. There are already several countries where there is a high percentage of people aged 65 years old and over, including Germany, Italy and Greece. It should be noted that Poland is also struggling with the problem of an ageing society because older people (65+) in our country constitute over 15% of the general population [40,41,42]. Analyzing the report published by the Central Statistical Office (CSO) presenting the population forecast for 2008–2035, it can be seen that in 2035 it is estimated that people aged 65 and over will make up 23.2% of the population, people 75 years and older will make up 12.2% of the population, and people 85 and older will make up 3.1% of the population, relative to the population as a whole [41]. This situation will force changes not only in the socioeconomic sector, but it will also mobilize health care sector representatives to implement activities aimed at obtaining new information on the functioning of the human body and changes that take place in it over the years. The aim of the study was to provide information on the biochemical factors determining the reduction of physical, mental and emotional skills of the elderly and senile, which can be used in the process of planning preventive measures to improve the quality of life of older people.

Analyzing the available literature, it can be stated that the ageing process of the body is associated with a number of changes in the systems of the body, including the immune system (increased concentration of proinflammatory cytokines such as interleukin-6) and the hormonal system (such as a decrease in melatonin levels) [5,7].

During the conducted research, the multidisciplinary research team sought to verify the truthfulness of the research hypothesis, which assumes the existence of a relationship between melatonin and interleukin-6 and the aging process.

The analysis did not show any relationship between the level of melatonin and interleukin-6 versus functional, cognitive and emotional skills or degree of health behaviors intensity among respondents aged 60–75 years who needed medical attention. It should be emphasized that the study showed a relationship between melatonin levels among people aged 60–75 years old who were active in everyday life and did not require inpatient medical care. Higher levels of melatonin were associated with poorer cognitive skills. It should be noted that in this group, no relationship between the level of interleukin-6 and functional, cognitive and emotional skills or degree of health behaviors intensity was found.

In a group of people over 75 years old the situation was different. The higher the pro-inflammatory cytokine level (interleukin-6), the lower the functional skills in performing complex daily activities, while the higher the level of melatonin. The eating habits were also less proper in the group of people over 75 years old who required medical care.

In a group of people over 75 years old who were active and who did not require hospitalization or medical care, a higher level of interleukin-6 determined lower functional skills in performing complex daily activities. In turn, one can associate higher levels of melatonin with better functional skills and a greater tendency to optimism.

As the obtained results show, it is difficult to state unequivocally what effect melatonin and interleukin-6 levels have on the aging process. Therefore, further research and complex analyses are necessary.

In this study, the study group and the comparative group did not differ significantly in their levels of interleukin-6 and melatonin. This study did not confirm the scientific reports that higher IL-6 levels in “healthy” older people can be treated as a marker of subclinical disease syndromes, leading to disability within a few years [4,5,6,7,8,9]. It should be noted that further research is needed to address this issue on a much larger study sample.

Previous studies, both cross-sectional and longitudinal, showed that higher levels of inflammatory cytokines are associated with a decrease in cognitive function [43,44,45,46,47,48,49]. The relationship between higher levels of IL-6 and increased risk of dementia was presented in a meta-analysis [50]. However, this was not clearly demonstrated in another study [51]. It was similar to a population clinical study about ageing (Mayo Clinic Study of Aging, MCSA) and multivariate analyses. They examined the cross-sectional and longitudinal associations between baseline-measured IL-6, IL-10, and tumor necrosis factor (TNFα) levels in the blood and the ratio of IL-6/IL-10 with cognitive test performance and mild cognitive impairment. There were 1602 elderly people in the studied community (median age = 72.8). The IL-6 to IL-10 ratio is considered a marker of the function of the congenital immune system [52]. In our study, there was no relationship between IL-6 level and cognitive skills of elderly people (60–75 years old) and senile people (over 75 years old) in the study group and comparative group. The results presented in this article are therefore similar to the results obtained by other authors [51,52]. A possible explanation for our observed lack of association between the inflammatory cytokine and cognitive skills can be the fact that the test was carried out only once. In the longer term studies, inflammatory cytokines were associated with poor cognitive skills. It is still possible that high levels of inflammatory cytokines are associated with a decrease in cognitive/mental function (MMSE scale) over longer periods of study. It should be noted that the present study also showed no significant relationship between the level of IL-6 in the study and comparative groups (65–75 years), functional skills (Barthel scale scores and IADL scale scores), emotional skills (LOT-R scores) or the intensity of health behavior (results of IHB). However, in the group of senile people, there was a correlation between the pro-inflammatory cytokine levels and functional skills. It has been shown that the higher the IL-6 level, the lower the efficiency in performing complex daily activities such as cleaning and shopping, both in the study and comparative groups. It is worth noticing that chronic inflammation can affect the loss of physical skills in older people or significantly reduce them, which leads to disability in the field of daily activities. These results are consistent with the results of another study in which two-dimensional analysis showed significantly higher levels of IL-6 in people with lower results on the ADL scale which assesses basic daily activities [53]. Also, a study conducted among 1020 participants (aged 65 and over) showed a significant relationship between inflammation (high levels of IL-6, CRP and IL-1RA) versus poor physical performance and muscle strength [54]. With age, the level of circulating pro-inflammatory cytokine increases, which contributes to a decrease in muscle strength which can significantly affect the efficiency of the human body [55]. This shows that the assessment of inflammatory markers can become a key screening test to assess the functional capacity of elderly and senile people.

Sleep disorders are a common problem among older people. Insomnia can negatively affect the physical and mental state of older people, and thus negatively affect their functioning in everyday life. Circadian disturbances in melatonin secretion may contribute to accelerated ageing. Oxidative stress and inflammation predispose people to loss of muscle strength, and thus to functional limitations and disabilities [56,57]. It has been shown that, apart from the pineal gland, melatonin receptors occur in the central nervous system, retina, heart, blood vessels and cells of the immune system as well as in the stomach, pancreas and small intestine [58,59]. It must be said that the level of this hormone decreases with age, and its circadian rhythm disappears in the elderly [60].

In the present study, no relationship between the level of melatonin versus functional, cognitive and emotional skills or intensity of health behaviors in the group of elderly people residing in ZOL or DDOM requiring medical support was found. Similar results were obtained in another study which assessed urinary melatonin levels in relation to cognitive function, physical fitness and mortality among 2821 older men [61]. There was no relationship between this hormone and these variables. It should be noted that our study showed a relationship between melatonin levels in elderly people (60–75 years old) from the comparative group and cognitive skills (MMSE scores). Higher levels of this hormone correlated with worse cognitive skills. The results obtained are inconsistent with the results of the study conducted among 1105 elderly people, where higher melatonin levels were associated with a lower frequency of cognitive impairment and depressed mood [62]. Possible discrepancies may result from the fact that in the above study, melatonin levels were assessed in urine samples whereas in the present study we used a venous plasma test. A meta-analysis of three randomized controlled trials also showed a negligible effect of melatonin supplementation on cognitive function, which was assessed using the MMSE scale [63].

In the present study, the relationship between melatonin levels in the senile belonging to the study group and the results of the subscale of proper eating habits was noticed. The higher the level of hormone produced by the pineal gland, the less proper the eating habits in this group of older people were. It should also be added that we found a relationship between the level of melatonin and the functional and emotional skills of people who were more than 75 years old belonging to the comparative group (people were more active and used the services of CAS and KAS). Higher levels of the hormone correlated with greater functional skills (IADL scores, Barthel scale scores) and a greater propensity for optimism (higher LOT-R scores). These results are similar to other studies which demonstrated a relationship between physiological levels of melatonin and mood [62]. A higher level of the hormone produced by the pineal gland was associated with lower prevalence of depressed mood among older people.

This study has many strengths, including population sample, a study and a comparative group, cross-sectional study structure and a high, sensitive measurement of inflammatory cytokine and melatonin in the venous plasma. Although the research provided important information about cognitive, emotional and functional skills of older people from a large Polish urban agglomeration, it has several limitations. One of them is a relatively small sample size, which was not representative. The conclusions cannot be generalized to the general population of older people but only this study sample. The study used targeted selection of participants, but it did not use a program to calculate sample size. Moreover, the age and gender of study and comparative groups was not homogeneous which may be a bias. These elements have been included as limitations of the study. It is our hope that the obtained results and conclusions will encourage others to explore the issues of subclinical markers of disease syndromes and holistic geriatric assessment using scales and tests assessing cognitive, functional and emotional skills. We also hope this study will be used in the development and implementation of new, effective disease diagnostic methods for the elderly in order to improve medical care for this special group of patients.

The obtained results may contribute to the development of a standard geriatric assessment scheme, with particular emphasis on biochemical indicators of the aging process. In clinical practice this will allow for rapid diagnosis of age-related pathologies, implementation of preventive and educational actions, and holistic medical care for older people.

## 5. Conclusions

The results of the research indicate that:The inflammatory marker IL-6 in blood plasma may not be useful in determining cognitive and emotional skills impairment or the intensity of health behaviors among elderly people (60–75 years), both active and requiring constant medical care. It can be partially treated as a predictive factor of functional skills, especially for people over 75 years of age.Melatonin can be used in the process of recognizing cognitive skill impairment in elderly people (60–75 years of age) who do not require constant medical care. It can also be used to recognize the functional and emotional skill impairment in senile people who are active and do not require hospitalization or medical support.There is a need for further research among older people in more centers in Poland. It also seems extremely important to pay attention to cultural diversity and diet, as well as intensity and type of physical activity of the surveyed people in the future.

## Figures and Tables

**Table 1 ijerph-17-03623-t001:** The comparison of study participants in terms of sociodemographic variables.

Parameters		The Study Group (N = 66) n/%	The Comparative Group (N = 55) n/%	*p* *
Gender	Female	48/72.73	49/89.09	*p* = 0.044
	Male	18/27.27	6/10.91	
Age	60–65 years old	3/4.55	17/30.91	*p* < 0.001
	66–70 years old	13/19.70	16/29.09	
	71–75 years old	6/9.09	11/20.00	
	76–80 years old	15/22.73	3/5.45	
	81–85 years old	15/22.73	6/10.91	
	86–90 years old	11/16.67	2/3.64	
	Over 90 years old	3/4.55	0/0.00	
Places of residence	Village	5/7.58	5/9.09	*p* = 0.639
	Town (≤50 thousand people)	1/1.52	0/0.00	
	Town (50–100 thousand people)	3/4.55	5/9.09	
	Big city (>100 thousand people)	57/86.36	45/81.82	

* Chi-square test/Fisher’s exact test. The comparison of values of qualitative variables in the groups was made using the chi-square test (with Yates correction for 2 × 2 tables) or Fisher’s exact test where the low expected numbers appeared in the tables. The statistical value is *p*.

**Table 2 ijerph-17-03623-t002:** The comparison of study participants in terms of functional, cognitive and emotional skills and lifestyle using standardized research tools.

Research Tools		Study Group (N = 66)	Comparative Group (N = 55)	*p* *
**Barthel scale**	X ± SD	40.3 ± 29.39	98.09 ± 3.4	*p* < 0.001
	Me	35	100	
	quartiles	20–40	95–100	
IADL	X ± SD	14.71 ± 4.37	23.65 ± 1.02	*p* < 0.001
	Me	13	24	
	quartiles	12–17	24–24	
MMSE	X ± SD	22.74 ± 3.35	27.53 ± 2.69	*p* < 0.001
	Me	22	28	
	quartiles	20–25	26–30	
LOT-R	X ± SD	14.76 ± 5.31	15.82 ± 3.19	*p* = 0.362
	Me	14	15	
	quartiles	12–19	13–18	
Total score of IHB	X ± SD	86.03 ± 16.93	95.44 ± 12.82	*p* = 0.002
	Me	89.5	97	
	quartiles	75.5–98.75	88–103.5	
IHB: Proper eating habits	X ± SD	3.33 ± 0.96	3.87 ± 0.81	*p* = 0.002
	Me	3.5	4	
	quartiles	2.5–4	3.42–4.42	
IHB: Preventive behaviors	X ± SD	3.43 ± 0.8	4.15 ± 0.72	*p* < 0.001
	Me	3.5	4.33	
	quartiles	3–4	3.92–4.67	
IHB: Positive mental attitude	X ± SD	3.66 ± 0.89	4.04 ± 0.57	*p* = 0.038
	Me	3.83	4	
	quartiles	3.04–4.33	3.67–4.5	
IHB: Health practices	X ± SD	3.91 ± 0.7	3.85 ± 0.68	*p* = 0.472
	Me	4	4	
	quartiles	3.67–4.33	3.42–4.33	

*, Mann–Whitney U-test. X, mean; SD, standard deviation; Me, median; *p*, statistical value.

**Table 3 ijerph-17-03623-t003:** The comparison of study participants in terms of interleukin-6 and melatonin values.

Parameters		Study Group (N = 66)	Comparative Group (N = 55)	*p* *
Interleukin-6 [pg/L]	X ± SD	22.29 ± 36.52	17.07 ± 24.16	*p* = 0.284
	Me	9.86	8.24	
	quartiles	5.43–20.11	3.38–20.32	
Melatonin [ng/L]	X ± SD	436.85 ± 336.05	471.24 ± 364.67	*p* = 0.893
	Me	319.75	308.62	
	quartiles	225.48–543.09	212.28–675.26	

*, Mann–Whitney U-test; X, mean; SD, standard deviation; Me, median; *p*, statistical value.

**Table 4 ijerph-17-03623-t004:** Functional, cognitive and emotional skills and lifestyle of participants from the study and comparative group.

Research Tools/Parameters		Study Group (N = 66) n/%	Comparative Group (N = 55) n/%
Barthel scale	“Severe” health condition	20/30.30	0/0.00
	“Medium level” health condition	33/50.00	1/1.82
	“Mild” health condition	13/19.70	54/98.18
MMSE	No cognitive impairment	11/16.67	39/70.91
	Cognitive impairment without dementia	16/24.24	12/21.82
	Light degree of dementia	39/59.09	4/7.27
LOT-R	Low	22/33.33	8/14.55
	Medium	15/22.73	24/43.64
	High	29/43.94	23/41.82
IHB	Low	17/25.76	5/9.09
	Medium	16/24.24	13/23.64
	High	33/50.00	37/67.27

**Table 5 ijerph-17-03623-t005:** The relationship between the level of interleukin-6 and melatonin and the results obtained from standardized research tools among respondents aged 60–75 belonging to the study group (n = 66).

Research Tools	Correlation (Spearman Rank Coefficient)			
	Interleukin-6		Melatonin	
Barthel scale	0.292	*p* = 0.188	0.059	*p* = 0.805
IADL	0.245	*p* = 0.272	0.115	*p* = 0.631
MMSE	−0.142	*p* = 0.527	−0.166	*p* = 0.485
LOT-R	−0.113	*p* = 0.618	−0.186	*p* = 0.433
Total score of IHB	0.26	*p* = 0.242	−0.087	*p* = 0.716
IHB: Proper eating habits	0.147	*p* = 0.515	0.131	*p* = 0.581
IHB: Preventive behaviors	0.099	*p* = 0.66	−0.063	*p* = 0.793
IHB: Positive mental attitude	0.111	*p* = 0.623	−0.176	*p* = 0.459
IHB: Health practices	0.251	*p* = 0.259	−0.090	*p* = 0.705

**Table 6 ijerph-17-03623-t006:** The relationship between the level of interleukin-6 and melatonin and the results obtained from standardized research tools among respondents aged 60–75 belonging to the comparative group (n = 55).

Research Tools	Correlation (Spearman Rank Coefficient)			
	Interleukin-6		Melatonin	
Barthel scale	−0.124	*p* = 0.424	−0.205	*p* = 0.205
IADL	0.06	*p* = 0.697	−0.227	*p* = 0.160
MMSE	−0.089	*p* = 0.564	−0.328	*p* = 0.039
LOT-R	0.258	*p* = 0.091	−0.255	*p* = 0.113
Total score of IHB	−0.081	*p* = 0.599	−0.096	*p* = 0.558
IHB: Proper eating habits	−0.124	*p* = 0.424	−0.025	*p* = 0.876
IHB: Preventive behaviors	0.149	*p* = 0.333	−0.282	*p* = 0.077
IHB: Positive mental attitude	−0.048	*p* = 0.758	−0.011	*p* = 0.945
IHB: Health practices	−0.165	*p* = 0.285	−0.008	*p* = 0.960

**Table 7 ijerph-17-03623-t007:** The relationship between the level of interleukin-6 and melatonin and the results obtained from standardized research tools among respondents over 75 years old belonging to the study group (n = 66).

Research Tools	Correlation (Spearman Rank Coefficient)			
	Interleukin-6		Melatonin	
Barthel scale	−0.183	*p* = 0.235	−0.118	*p* = 0.463
IADL	−0.331	*p* = 0.028	−0.042	*p* = 0.795
MMSE	−0.202	*p* = 0.187	−0.275	*p* = 0.082
LOT-R	−0.154	*p* = 0.319	−0.014	*p* = 0.929
Total score of IHB	−0.033	*p* = 0.832	−0.276	*p* = 0.081
IHB: Proper eating habits	0.085	*p* = 0.582	−0.373	*p* = 0.016
IHB: Preventive behaviors	−0.027	*p* = 0.863	−0.246	*p* = 0.121
IHB: Positive mental attitude	−0.108	*p* = 0.485	−0.054	*p* = 0.738
IHB: Health practices	−0.082	*p* = 0.596	−0.146	*p* = 0.363

**Table 8 ijerph-17-03623-t008:** The relationship between the level of interleukin-6 and melatonin and the results obtained from standardized research tools among respondents over 75 years old belonging to the comparative group (n = 55).

Research Tools	Correlation (Spearman Rank Coefficient)			
	Interleukin-6		Melatonin	
Barthel scale	−0.306	*p* = 0.359	0.663	*p* = 0.026
IADL	−0.724	*p* = 0.012	0.613	*p* = 0.045
MMSE	0.433	*p* = 0.183	0.284	*p* = 0.398
LOT-R	−0.035	*p* = 0.919	0.651	*p* = 0.030
Total score of IHB	−0.174	*p* = 0.610	0.064	*p* = 0.852
IHB: Proper eating habits	−0.461	*p* = 0.154	−0.181	*p* = 0.594
IHB: Preventive behaviors	−0.253	*p* = 0.452	−0.110	*p* = 0.747
IHB: Positive mental attitude	0.266	*p* = 0.429	0.476	*p* = 0.139
IHB: Health practices	−0.169	*p* = 0.619	0.224	*p* = 0.508

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
