# Peer review of "Interleukin-6 and Melatonin as Predictors of Cognitive, Emotional and Functional Ageing of Older People"

_ijerph, 2020, doi:10.3390/ijerph17103623_

Round 1

Reviewer 1 Report

11-space after colon

23 "however" does not fit.

28-30 Unnecessary generalization. Cut.

31, English problem.  Cognitive, emotional and functional skills are three factors.

Better: "Some main factors that affect independence includes cognitive, emotional and functional skills."

40-43, Three "it is also" in a roll.

The problems are stylistic and grammatical, e.g. excessive passive voice usage, inconsistency of tense, etc.  I will stop correcting the English here, but recommend careful revision and editing.

270, I think the author means "the rapidly ageing population," not "rapid ageing process of the societies."

Question:

According to the methodology, the research team had conducted interviews with the participants.  Would the author provide a more in depth discussion concerning how the results of the interviews affect the interpretation of the biochemical data, particularly concerning the context of health of any given individual participant in relation to the biochemical results (if such information is available)? 

Author Response

Author’s Notes to Reviwer

Reviwer 1:

Thank you for your review and valuable remarks and tips that have been included in the study.

  1. 11-space after colon

Response: Thank you for this comment. Correction made.

  1. 23 "however" does not fit.

Response: Thank you for this suggestion. Correction made.

  1. 28-30 Unnecessary generalization. Cut.

Response: Thank you for this comment. The authors deleted the indicated sentence in the text.

  1. 31, English problem. Cognitive, emotional and functional skills are three factors.

Better: "Some main factors that affect independence includes cognitive, emotional and functional skills."

Response: Thank you for this suggestion. Correction made. The sentence has been corrected as suggested by the Reviewer #1.

  1. 40-43, Three "it is also" in a roll.

The problems are stylistic and grammatical, e.g. excessive passive voice usage, inconsistency of tense, etc.  I will stop correcting the English here, but recommend careful revision and editing.

Response: Thank you for this assessment. We have reanalysed the content of the article and corrected the text taking into account the Reviewer’s suggestions about stylistic and grammatical problems in this article.

  1. 270, I think the author means "the rapidly ageing population," not "rapid ageing process of the societies."

Response: Thank you for this comment. We have corrected the sentence taking into account the Reviewer’s suggestions.

  1. Question: According to the methodology, the research team had conducted interviews with the participants. Would the author provide a more in depth discussion concerning how the results of the interviews affect the interpretation of the biochemical data, particularly concerning the context of health of any given individual participant in relation to the biochemical results (if such information is available)?

Response: Thank you for these valuable suggestions. We have added more detailed information in the “Methods, Techniques and Research Tools” section and we have clarified the Reviewer’s #1 doubts.

During the study, various methods were used to collect data, including the diagnostic survey method and interviews. The study participant independently filled in three research tools: IADL, LOT-R and the brief demographic questionnaire. While completing the other two standardized research tools - MMSE and the Barthel scale, a member of the research team was present and asked questions to the subject (interviewed). The MMSE and the Barthel scale are tools completed by a person with medical/ psychological education. The participant cannot objectively fill these tools in on their own, which explains why the interview appeared in the study, which collected information about cognitive skills (MMSE) and functional skills (Barthel scale) of people from the study and comparative group.

The relation between the results collected during interview (MMSE and Barthel scale results) and biochemical results (melatonin and interleukin 6) were presented in Tables 5 and 6.

Reviewer 2 Report

Firstly, I congratulate to the authors! I consider that they have done a great work! I found this paper particularly interesting because the population is aging, and the authors evaluated the relationship between cognitive, emotional, functional skills and lifestyle 65 with melatonin and interleukin 6 (IL-6) in older people.

However, I consider that this paper can be improved by grounding it in a specific problem/hypothesis and edit in depth the English grammar. This would give the paper much more purpose as well as creating the basis for a more substantial discussion, which at the present time lacks structure.

I appreciate you will have put in a lot of effort in preparing your paper and there are several points which I offer to assist with further developing the manuscript. These points are listed under the subheadings.

Introduction:

  1. Try to eliminate the use of 'elderly' and use 'older people' or “older persons” because these is considered more appropriate
  2. Hypotheses and aims should be included in this section.
  3. What it is the meaning of senile people? What is the different between elderly and senile people? This should be explained!

Methods:

  1. The aims should be moved to the end of introduction section.
  2. The “Organisation and Course of the Study” section could be divided in three section: study design, procedure and Ethical Considerations.
  3. Why are there two groups? It is not clear to me if it is a cross-sectional or case-control study.
  4. Did you do sampling? Did you used a program to calculate sample size? What was the reference population? Please, provide more information about it.
  5. I recommend use “people over 75 years” instead of “senile people”.
  6. The age and gender of study and comparative groups it is not homogeneous. Why? This may be a bias.
  7. Did the Barthel and Lawton scales adapted to Polish population?
  8. You use "senior" to refer to older people in the different scales and this confuses the reader. I recommend that you use older people in all document due to you use many terms: elderly, older people, seniors, senile people, etc.
  9. On page 5, line 171 you say “The results were interpreted with an immunoenzymatic markings machine by the use of specialized software” What was the software? Explain it!

Discussion:

  1. A very brief summary of the main results is missing in the beginning of the discussion.
  2. Discussion in this section is fragmented due to the paper not being embedded into a specific problem/hypothesis.
  3. This section could be structured better. The limitations of the study are missing.
  4. It would be interesting that the authors explain better the implications of this study for clinical practice.

Conclusion:

  1. Could you identify future directions?

References:

  1. I think that your literature should be the most recent on this topic; there are only 12 of 59 references that belong to last five years. Try to include and change by the most up to date international literature!

General:

  1. When you refer to a study, it is not necessary to put the name of the authors and published journal. The citation is enough!
  2. There are several grammatical and styling errors, for example: in the title appears “funtional” instead of “functional”. The article must be edited in English in depth for their correct compression.

Author Response

Author’s Notes to Reviwer

Reviwer 2:

Thank you for your review and valuable remarks and tips that have been included in the study.

  1. Firstly, I congratulate to the authors! I consider that they have done a great work! I found this paper particularly interesting because the population is aging, and the authors evaluated the relationship between cognitive, emotional, functional skills and lifestyle 65 with melatonin and interleukin 6 (IL-6) in older people.

Response: Thank you for this positive assessment.

  1. However, I consider that this paper can be improved by grounding it in a specific problem/hypothesis and edit in depth the English grammar. This would give the paper much more purpose as well as creating the basis for a more substantial discussion, which at the present time lacks structure.

I appreciate you will have put in a lot of effort in preparing your paper and there are several points which I offer to assist with further developing the manuscript. These points are listed under the subheadings.

Response: We appreciate this comment. We have reanalysed the content of the article and corrected the text taking into account the Reviewer’s suggestions.

  1. Introduction:

Try to eliminate the use of 'elderly' and use 'older people' or “older persons” because these is considered more appropriate

Hypotheses and aims should be included in this section.

What it is the meaning of senile people? What is the different between elderly and senile people? This should be explained!

Response: Thank you for these valuable suggestions. We have reanalysed the content of the article and corrected the title of the article (we eliminated “elderly and senile people” and used “older people”). The aims of the study have been moved to the ‘Introduction’ section. Moreover, we have added hypothesis.

The difference between elderly and senile people was explained in the “Methods, Techniques and Research Tools” section by the authors of the study. As suggested by the Reviewer #2, this information has been moved to the “Introduction” section.

Based on the division of individual stages of old age proposed by the World Health Organization (WHO), respondents were assigned in this study to two age groups: 60-75 years old – early old age (elderly people) and 76-90 years old which is called the period of old age (senile people).

  1. Methods:

The aims should be moved to the end of introduction section.

Response: Thank you for this comment. The aims of the study have been moved to the “Introduction” section taking into account the Reviewer’s suggestions

  1. The “Organisation and Course of the Study” section could be divided in three section: study design, procedure and Ethical Considerations.

Response: We appreciate this comment. As suggested by the Reviewer #1, were analised the content of the “Organisation and Course of the Study” section and we divided it in three sections: “Study design”, “Procedure”, “Ethical Considerations”.

  1. Why are there two groups? It is not clear to me if it is a cross-sectional or case-control study.

Response: Thank you for this comment. The article presents a cross-sectional study. The respondents were examined at one point in time (information about variables was collected at the same time using questionnaires or an interviews). This study is  a descriptive study (neither longitudinal nor experimental). At the same time, melatonin and interleukin 6 levels as well as cognitive, physical and emotional skills were tested. Although the study included the study group (hospitalized) and the comparative group (people who did not use medical care on a regular basis), the study was not prospective or retrospective. A cause and effect relationship between exposure and disease has not been established during the study. The authors of this work introduced two groups to make the research more interesting and varied.

  1. Did you do sampling? Did you used a program to calculate sample size? What was the reference population? Please, provide more information about it.

Response: We appreciate this comment. A group of 121 people participated in the study. Targeted selection of participants was used in the study. The programme to calculate sample size was not used during the study. These elements were included as limitations (of the study) in the “Discussion” section. The study was voluntary, the patients decided themselves  whether they wanted to participate.

We have created new “Sampling” section in the article and added more detailed information about reference population. Data from the Central Statistical Office of Poland indicate that the number of old people (65+) in the lesser Poland voivodship in 2018 was 680 178 people (Central Statistical Office, 2020). Therefore, respondents participating in the study constituted 0.02% of the reference population. Analyzing the data only for the city of Krakow (from which over 80% of respondents came from) it can be seen that in 2018 the number of senile people registered in this city was 183 036. Therefore, the group participating in the survey constituted approximately 0.07% of the population of people over 65 in Krakow (BIP, 2020).

Local Data Bank. Population status. Central Statistical Office (CSO). https://bdl.stat.gov.pl/BDL/dane/podgrup/tablica (accessed on 4th May 2020).

Public Information Bulletin of Cracow (BIP). https://www.bip.krakow.pl/ (accessed on 4th May 2020).

  1. I recommend use “people over 75 years” instead of “senile people”.

Response: We appreciate this comment. Based on the division of individual stages of old age proposed by the World Health Organization (WHO), respondents were assigned in this study to two age groups: 60-75 years old – early old age (elderly people) and 76-90 years old which is called the period of old age (senile people).

The difference between elderly and senile people was explained in the “Introduction” section which makes easier to analyse the further content of the article.

  1. The age and gender of study and comparative groups it is not homogeneous. Why? This may be a bias.

Response: We agree with the Reviewer  #2 that the age and gender of study and comparative groups is not homogeneous. This may be a bias and it has been included as limitation of the study in the “Discussion” section.

  1. Did the Barthel and Lawton scales adapted to Polish population?

Response: Thank you for this comment. Unfortunately, the Barthel and Lawton scale has not been officially adapted to Polish population. It should be noted that the Barthel Scale in Poland is the basis for referring patients to Care and Treatment Institution and long-term nursing care facilities in the place of residence in accordance with the Regulation of the Minister of Health of 22nd November 2013. The Barhel and Lawton scales are commonly used in Poland during the comprehensive geriatric assessment  of older people.

  1. You use "senior" to refer to older people in the different scales and this confuses the reader. I recommend that you use older people in all document due to you use many terms: elderly, older people, seniors, senile people, etc.

Response: We appreciate this comment. Thank you for these valuable suggestion. We have reanalysed the content of the article and corrected the text. We eliminate “senior” and use “older people”.

During analysing and presenting the results, we use the concept of “elderly people“ and “senile people”, which was explained at the beginning of the article in the “Introduction” section.

  1. On page 5, line 171 you say “The results were interpreted with an immunoenzymatic markings machine by the use of specialized software” What was the software? Explain it!

Response: Thank you for this comment. We have corrected the text in “Biochemical Blood Test” section  to make it clearer. Blood tests were performed by enzyme immunoassay method. We cannot provide the names of the laboratory devices used, because this may involve the disclosure of their trade names and the disclosure of laboratory equipment. This could be an element that suggests a conflict of interest.

  1. Discussion:

A very brief summary of the main results is missing in the beginning of the discussion.

Discussion in this section is fragmented due to the paper not being embedded into a specific problem/hypothesis.

This section could be structured better. The limitations of the study are missing.

Response: Thank you for these valuable suggestions. We have modified the “Discussion” section.  We have added brief summary of the main results at the beginning of this section and information about the limitations of the study.

During the conducted research, the multidisciplinary research team sought to verify the truthfulness of the research hypothesis, which assumes the existence of a relationship between melatonin and interleukin 6 and the aging process.

The analysis did not show any relationship between the level of melatonin and interleukin 6 versus functional, cognitive, emotional skills and the degree of health behaviours intensity among respondents aged 60-75 years who needed medical attention. It should be emphasized that the study showed a relationship between melatonin levels among people aged 60-75 years old who were active in everyday life and did not require inpatient medical care, and between cognitive skills. Higher levels of melatonin were associated with poorer cognitive skills of these people. It should be noted that in this group no the relationship between the level of interleukin 6 and functional, cognitive, emotional skills and the degree of health behaviours intensity was found.

In a group of people over 75 years old the situation was different. The higher the pro-inflammatory cytokine level (interleukin 6), the lower the functional skills in performing complex daily activities were, while the higher the level of melatonin was, the less proper eating habits were in the group of people over 75 years old who required medical care.   

In a group of people over 75 years old who were active and who did not require hospitalization or medical care, a higher level of interleukin 6 determined lower functional skills in performing complex daily activities. In turn, one can associate higher levels of melatonin with better functional skills and a greater tendency to optimism.

As the obtained results show, it is difficult to state unequivocally what effect melatonin and interleukin 6 levels have on the aging process, therefore further research and complex analyses are necessary.

Although the research provided important information about cognitive, emotional and functional skills of the older people from the area of a large Polish urban agglomeration, which belongs to a large region of Lesser Poland, it has several limitations. One of them is a relatively small sample size, which was not representative. The conclusions cannot be generalized to general population of older people but only study sample. The study used targeted selection of participants but it did not use a programme to calculate sample size. Moreover, the age and gender of study and comparative groups were not homogeneous. This may be a bias. These elements have been included as limitation of the study.  

  1. It would be interesting that the authors explain better the implications of this study for clinical practice.

Response: We appreciate this comment and we have added more detailed information about implications of this study for clinical practice in the “Discussion” section. The obtained results may contribute to the development of a standard geriatric assessment scheme, with particular emphasis on biochemical indicators of the aging process, which in clinical practice will allow rapid diagnosis of age-related pathologies, implementation of preventive and educational actions, and providing holistic medical care for older people.

  1. Conclusion:

Could you identify future directions?

Response: Thank you for these suggestions. The results indicate the need for further research among older people in more centres  in Poland. It also seems extremely important to pay attention to the cultural diversity, diet and the intensity and type of physical activity of the surveyed old people in the future.

  1. References:

I think that your literature should be the most recent on this topic; there are only 12 of 59 references that belong to last five years. Try to include and change by the most up to date international literature!

Response: Thank you for this comment. The authors of this article had difficulty finding current publications that related to the presented issue. They have reanalysed the content of the References and introduced several new items (6 references) in the literature in such a way  as not to disturb the entire structure of the article.

  1. General:

When you refer to a study, it is not necessary to put the name of the authors and published journal. The citation is enough!

Response: Thank you for these suggestions. Correction made.

  1. There are several grammatical and styling errors, for example: in the title appears “funtional” instead of “functional”. The article must be edited in English in depth for their correct compression.

Response: Thank you for this comment . We have reanalysed the content of the article and corrected the text taking into account the Reviewer’s suggestions.

Round 2

Reviewer 2 Report

I'm pleased you have amended the paper which is more much more readable. The authors did a great job in revising the manuscript and thoroughly addressed reviewer comments. I have a few suggestions which are relatively minor but may help improve the manuscript:

  1. In introduction section, page 2 line 61-61, the authors twice repeat the phrase “based on the available literature”.
  2. It is not clear to me how the participants were selected. Were they selected for convenience or through the user lists? Please, explain it!

I hope these suggestions assist to improve this manuscript that I consider it is very interesting. Best wishes.

Author Response

I'm pleased you have amended the paper which is more much more readable. The authors did a great job in revising the manuscript and thoroughly addressed reviewer comments. I have a few suggestions which are relatively minor but may help improve the manuscript:

Thank you for this positive assessment.

In introduction section, page 2 line 61-61, the authors twice repeat the phrase “based on the available literature”.

Thank you for this comment. Correction made.

It is not clear to me how the participants were selected. Were they selected for convenience or through the user lists? Please, explain it!

Thank you for these valuable suggestion. Participants were not selected for convenience or through the user lists. Targeted selection of participants was used in the study. In consultation with the managers of facilities potential participants (meeting the inclusion criteria) were initially selected. The main criterion was informed consent to participate in the study, which was associated with absence of mental illness, Alzheimer's disease or other severe neurodegenerative diseases which make it difficult to make an informed decision. The study was voluntary, the patients decided themselves whether they wanted to participate. Targeted selection of participants was included as limitations (of the study) in the “Discussion” section.

We have added more detailed information about it in “Sampling” section.

I hope these suggestions assist to improve this manuscript that I consider it is very interesting. Best wishes.

Thank you for these valuable suggestions and positive assessment of the manuscript.